# A Subpixel Residual U-Net and Feature Fusion Preprocessing for Retinal Vessel Segmentation

**Abstract.** Retinal Image analysis allows medical professionals to inspect the morphology of the retinal vessels for the diagnosis of vascular diseases. Automated extraction of the vessels is vital for computer-aided diagnostic systems to provide a speedy and precise diagnosis. This paper introduces SpruNet, a Subpixel Convolution based Residual U-Net architecture which re-purposes subpixel convolutions as down-sampling and up-sampling method. The proposed subpixel convolution based down-sampling and up-sampling strategy efficiently minimize the information loss during the encoding and decoding process which in turn increases the sensitivity of the model without hurting the specificity. A feature fusion technique of combining two types of image enhancement algorithms is also introduced. The model is trained and evaluated on three mainstream public benchmark datasets, and detailed analysis and comparison of the results are provided which shows that the model achieves state-of-the art results with less complexity. The model can make inference on 512x512 pixel full image in half of a second. The code is available at: https://link is hidden for anonymity

**Keywords:** Medical Image Analysis, Retinal Vessel Segmentation, Sub-pixel Convolution, Residual Network, U-Net

## 1 Introduction

Retinal Vessel Analysis is a non-invasive method of examining the retinal vasculature comprising of a complicated and elaborate network of arteries, veins and capillaries. The morphology of the retinal vasculature is an important biomarker for diseases like Diabetic retinopathy, Hypertensive retinopathy, Retinal vein occlusion, Retinal artery occlusion, etc., which affects the retinal blood vessels. Retinal examination allows medical professionals dealing with vascular diseases to get a unique perspective, allowing them to directly inspect the morphology and draw conclusions about the health of a patient's microvasculature anywhere in the body. The retinal vasculature is also adopted as the most stable feature for multimodal retinal image registration and retinal mosaic. They are also being used for biometric identification.

Quantitative analysis of the retinal blood vessels, requires the vascular tree to be extracted so that morphological features like length, width, branching, angle, etc. can be calculated. Manual segmentation of the blood vessel is a difficult

and time consuming task and requires expertise. Automating the task of vessel segmentation has gained importance and has been accepted as a vital and challenging step for retinal image analysis in computer aided diagnostic systems for ophthalmic and cardiovascular diseases.

Retinal vessels vary in shape, size and intensity level locally. The vessel width may range anywhere between 1 to 20 pixels depending on the anatomical vessel width and image resolution. Segmentation using artificially designed features or conventional image-processing based segmentation algorithms is quite difficult because of the presence of vessel crossing, overlap, branching and centerline reflex. The Segmentation can be further complicated due to the presence of pathologies in the form of lesions and exudates.

Deep learning based supervised segmentation models have proved to perform much better than the classical unsupervised methods which depend on hand crafted features. While most of them uses an encoder-decoder architecture, the image is down-sampled and up-sampled many times in the encoder and decoder respectively. These two processes are most commonly performed by a max-pooling operation and transposed convolution operation respectively and in the process, some amount of important information or features are lost which could have been beneficial for the segmentation algorithm. This paper introduces a Subpixel Residual U-Net architecture or SpruNet, based on the U-Net [11] framework, which re-purposes the subpixel convolutions [13] to perform image down-sampling and up-sampling in the U-shaped encoder-decoder architecture. This approach preserves information and provides better accuracy than the state-of-the-art algorithms. Also, this architecture is simpler, faster and has fewer parameters ($\sim$20M) than the previous state-of-the-art algorithms.

The main problem with retinal images is the varying brightness and contrasts. To tackle this, a feature fusion technique is proposed to increase the accuracy and robustness of the model a bit more. Contrast Limited adaptive histogram equalization(CLAHE) [10] is combined with Ben Graham's [2] pre-procesing method of subtracting the local average colour. This increases the clarity of the vessels in most of the images as compared to using either of the algorithms alone. But after some experiments its seen that in some scenarios, specially on the Chase dataset the combination of CLAHE and Ben Graham's algorithm performs slightly worse than the standalone CLAHE. To solve this, a feature fusion approach of concatenating the standalone CLAHE preprocessed image with the combined CLAHE and Ben Graham's preprocessed image is used, which tops all of the experiments done on all three datasets.

## 2   Related Work

In the last couple of years significant improvement has been seen in the field of Retinal Image Analysis, especially for the task of vessel extraction. This section gives a brief overview of the latest high-performance supervised approaches.

Roychowdhury S. et al. [12] in 2015 proposed a unsupervised method for Retinal Vessel Segmentation where they used iterative adaptive thresholding to

indentify vessel pixels from vessel enhanced image which is generated by tophat reconstruction of the negative green plane image.

Liskowski P. and Krawiec K. [7] in 2016 proposed a supervised deep neural network model for Retinal Vessel Segmentation and preprocessed the images with global contrast normalization, zero-phase whitening, and used different augmentations like geometric transformations and gamma corrections.

Orlando J.I. et al. [9] in 2017 proposed a conditional random field model for Retinal Vessel Segmentation which enables real time inferencce of fully connected models. It uses a structured output support vector machine to learn the parameters of the method automatically.

Alom M.Z. et al. [1] in 2018 proposed their R2Unet which harnessed the power of Recurrent Convolutional Neural Networks (RCNN). The residual blocks allowed the network to train faster while the recurrent convolution layers allowed feature accumulation which helped in better feature representation.

Wang B. et al. [14] in 2019 proposed their Dual Encoding U-Net (DEU-Net) architecture, having two encoding paths: a spatial path with large kernels and a context path with multi-scale convolution blocks. This allowed the network to preserve the spatial information and encode contextual information via the two pathways.

Wu Y. et al. [3] in 2019 proposed their Vessel-Net, where they used inception inspired residual convolutional blocks in the encoder part of a U-like encoder-decoder architecture and introduced four supervision paths to preserve the rich and multi-scale deep features.

Jin Q. et al. [5] in 2019 proposed their DUNet architecture based on the U-Net which used deformable convolution blocks in place of few of the standard convolutional blocks. Deformable convolution blocks allow the network to adaptively adjust the receptive fields, thus enabling this architecture to classify the retinal vessels at various scales.

Yan Z. et al. [15] in 2019 proposed a three stage model for Retinal Vessel Segmentation, which segments thick vessels and thin vessels separately. The separate segmentation approach helps in learning better discriminative features. The final stage refines the results and identifies the non vessel pixels. This way the overall vessel thickness consistency is improved

Zang S. et al. [16] in 2019 proposed their Attention Guided Network for Retinal Vessel Segmentation which uses a guided filter as a structure sensitive expanding path and an attention block which exclude noise. Their method preserves structural information.

Li L. et al. [6] in 2020 proposed their IterNet. They used a standard U-Net followed by an iteration of mini U-Nets with weight-sharing and skip connections. This allowed them to pass the output features of the standard U-Net through a number of light-weight intermediate networks to fix any kind of defects in the results.

## 3    Proposed Method

### 3.1   Dataset

The following three prominent benchmark datasets are used for experiment:

- The **Drive** dataset has 40 colored images of the retinal fundus along with the pixel-level segmentation mask. The size of the images is 584x565 pixels. The dataset already has a predefined 20-20 train-test split.
- The **Chase** dataset has 28 colored images of the retinal fundus along with the pixel-level segmentation mask. The size of the images is 999x960 pixels. The dataset has no predefined split so a random 20-8 train-test split is used.
- The **Stare** dataset has 20 colored images of the retinal fundus along with the pixel-level segmentation mask. The size of the images is 700x605 pixels. The dataset has no predefined split so a random 16-4 train-test split is used.

### 3.2   Preprocessing and Augmentation

The retinal images come with different lighting conditions and vary a lot in terms of brightness and contrast. Also different methods of image acquisition and presence of diseases makes it hard to build a robust segmentation algorithm. Image enhancement algorithms are used to increase the clarity of the images as much as possible before passing them to the model. The most popular retinal image enhancement algorithm used in the literature is the Contrast Limited Adaptive Histogram Equalization (CLAHE) algorithm. It increases the contrast of local regions to enhance the visibility of local details. It is quite robust and performs very well in most scenarios. In 2015 Ben Graham used a preprocessing technique in the Kaggle Diabetic Retinopathy competition, where he subtracted the local average colour from the image. In this method the vessels popped out quite distinctively from the background. In this paper we further enhance this method with a two step preprocessing. First CLAHE is used to increase the contrast, and then the local average colour is subtracted out from it, resulting in better clarity as shown in Fig. 1.

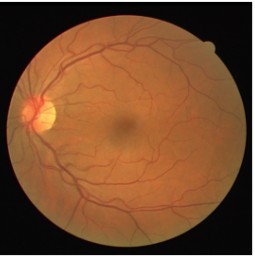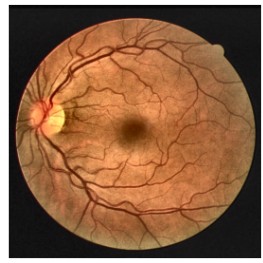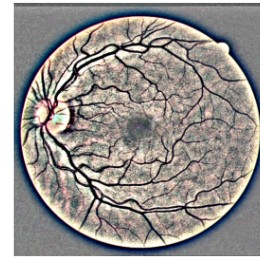

**Fig. 1.** From left to right: Original Images, CLAHE, CLAHE+Local Mean Colour Subtraction

Though this method performed better than the standalone CLAHE method on both the Drive and Stare dataset, it performed slightly worse on the Chase dataset. To tackle this problem, a feature fusion approach is taken, and a three step preprocessing is done as explained in the following steps.

1. Perform CLAHE on original image.
2. Perform Ben Graham's preprocessing of subtracting the local average colour from the CLAHE preprocessed image from step 1.
3. Concatenate the preprocessed images from step 1 and step 2 along the channels.

This method allows the model to use the all the information available in the CLAHE preprocessed image from step 1 along with the local average colour subtracted image from step 2. A diagram is shown in Fig. 2. A detailed ablation study is shown later which shows the improvement achieved with the Feature Fusion preprocessing method.

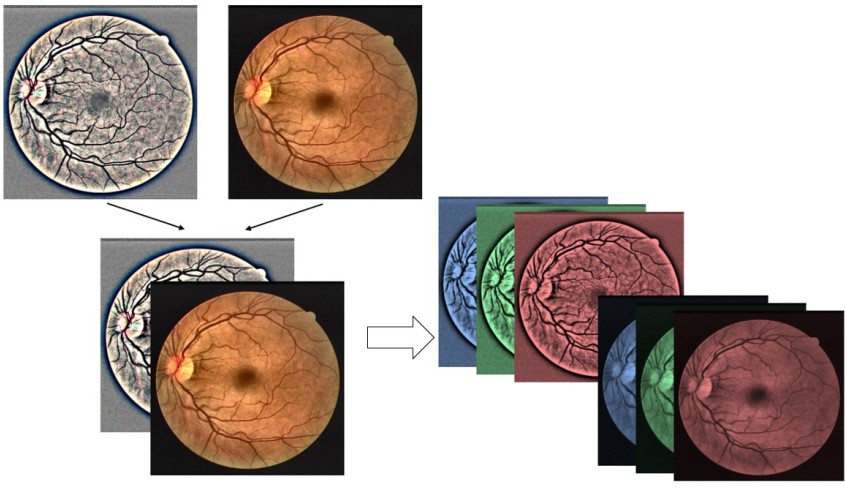

**Fig. 2.** Concatenating the preprocessed images

There are only a few public retinal datasets which includes vessel segmentation ground truths and all of them contains very few images. Since only around 20 images from each of the above-mentioned datasets are used for training, heavy data augmentation is used to increment the sample count to avoid overfitting. This includes: random horizontal and vertical flips, random rotations, minor RGB shifts for few images, and random brightness, contrast and gamma. Spatial level transformation like Grid distortion, Elastic transform and Optical distortion is also used to prevent over-fitting.

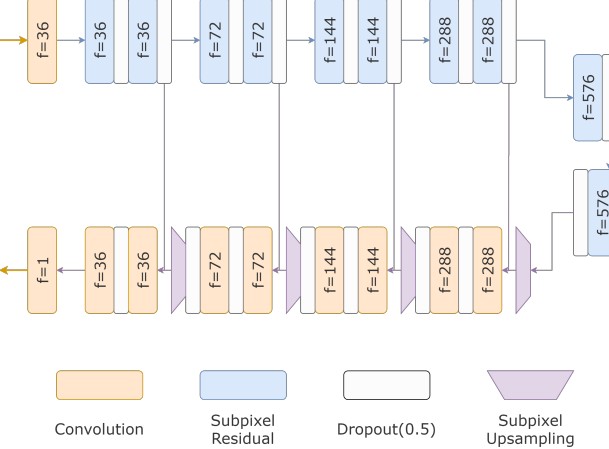

Convolution          Subpixel Residual          Dropout(0.5)          Subpixel Upsampling

**Fig. 3.** Architecture

## 3.3   Architecture

The proposed architecture - SpruNet or Subpixel Residual U-Net (shown in Fig. 3) is adapted from the encoder-decoder architecture of the U-Net. The model is a fully convolutional model [8] with a contraction and expansion path. The encoder is a 25-layer novel ResNet [4] architecture which uses subpixel residual blocks instead of standard residual convolution blocks. The decoder is a stack of 13-layers with two convolutional layers following every subpixel convolutional up-sampling block. Skip connections from encoder to decoder facilitates feature transfer from earlier layers to later layers which helps in the fine-grained segmentation. Each convolution layer is followed by a batch-normalization layer. Relu activation is used in all the layers except in the convolution layer in the subpixel convolution block.

   Subpixel Convolution is re-purposed as a down-sampling and up-sampling method for semantic segmentation in the encoder and decoder respectively. Subpixel Convolution is just a standard convolution followed by a pixel reshuffle. Normally a Subpixel convolution is used for up-sampling process, but we adapt it to be used for both tasks as it preserves the data unlike any other methods commonly used. Fig. 4 shows how the image dimension changes in the subpixel down-sampling block and the subpixel up-sampling block. Fig. 5 shows how the residual convolution block is adapted to use subpixel convolution.

– **Subpixel down-sampling**: In this the image of dimension (H, W, C) is converted to (H/2, W/2, 2C). For this we first pass it through a 1x1 2D convolution layer to reduce the depth to (H, W, C/2), then use pixel shuffling to change the dimension from (H, W, C/2) to (H/2, W/2, 2C). This process proves to be more efficient in preserving the spatial information as we down-sample the image but at the same time encode sufficient semantic information for efficient pixel-wise classification.

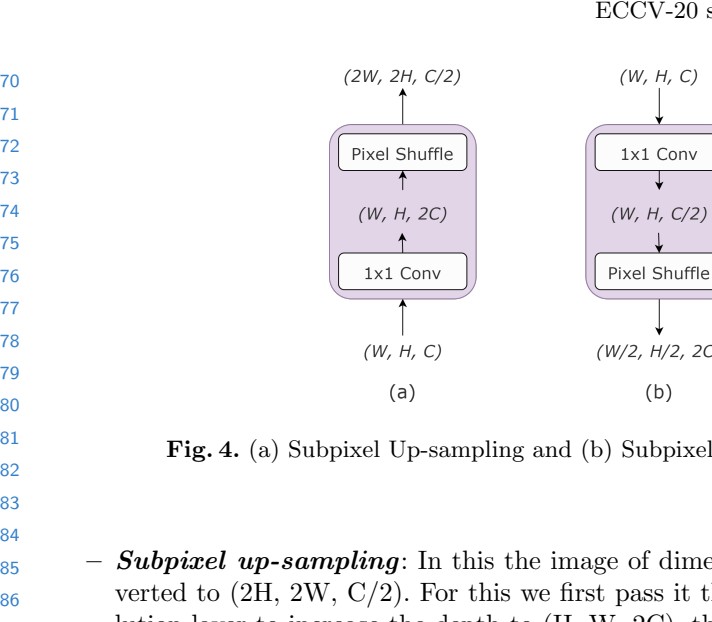

**Fig. 4.** (a) Subpixel Up-sampling and (b) Subpixel Down-sampling

– **_Subpixel up-sampling_**: In this the image of dimension (H, W, C) is converted to (2H, 2W, C/2). For this we first pass it through a 1x1 2D convolution layer to increase the depth to (H, W, 2C), then use pixel shuffling to change the dimension from (H, W, 2C) to (2H, 2W, C/2).

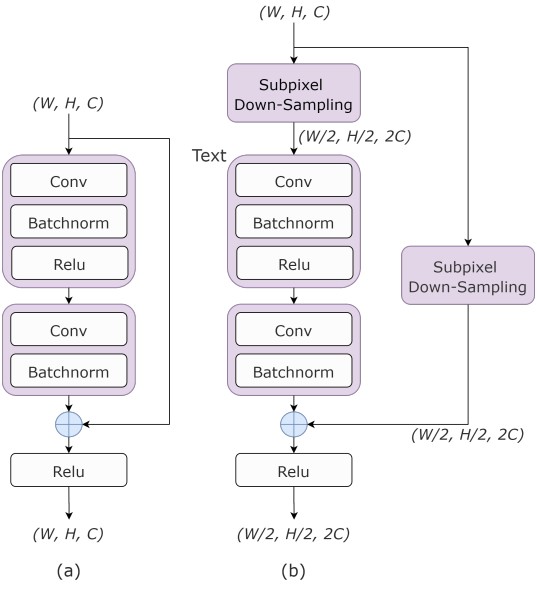

**Fig. 5.** (a) Identity Block and (b) Subpixel Residual Block

### 3.4   Loss Function

We used BCE-Dice loss, a combination of pixel-wise Binary Cross-entropy loss which compares each pixel individually and Dice loss which measures the amount of overlap between two objects in an image. Pixel-wise cross-entropy loss suffers from class imbalance while the Dice loss has a normalizing effect and is not affected by class imbalance. This combination gave better segmentation accuracy than any one of them used individually.

$$BCE\text{-}Loss = y(log(p) + (1 - y)log(1 - p) \tag{1}$$

$$Dice\text{-}Loss = 2 \cdot \frac{A \cap B}{A \cup B} \tag{2}$$

$$BCE\text{-}Dice\text{-}Loss = BCE\text{-}Loss + Dice\text{-}Loss \tag{3}$$

### 3.5   Metrics

We evaluated our model on 5 evaluation metrics to provide a good comparison with the other methods: Accuracy, F1-score, Sensitivity, Specificity and ROC-AUC. The F1 score is the harmonic mean of the precision and recall. Since there is a large class imbalance, the F1 score is a better metric than the Accuracy. The higher the F1 score the better. Sensitivity is the ability of the model to correctly identify vessel pixels (true positive rate), whereas Specificity is the ability of the model to correctly identify non-vessel pixels (true negative rate). ROC - curve is a graph which plots the true positive rate against the true negative rate at various thresholds. AUC is the area under the ROC curve. The higher the AUC the better the model is at distinguishing vessel vs non-vesel pixels.

$$F1\text{-}score = 2 \cdot \frac{Precision \cdot Recall}{Precision + Recall} \tag{4}$$

$$Sensitivity = \frac{TP}{TP + FN} \tag{5}$$

$$Specificity = \frac{TN}{TN + FP} \tag{6}$$

**Table 1.** Ablation table for Drive dataset

| Method | F-1 | SE | SP | AC | AUC |
|---|---|---|---|---|---|
| CLAHE + Subpixel | 0.8445 | 0.8296 | 0.9843 | 0.9682 | 0.9868 |
| CLAHE + Ben Graham's + Subpixel | 0.8514 | 0.8394 | 0.9848 | 0.9701 | 0.9874 |
| Information Fusion + No-Subpixel | 0.8477 | 0.8287 | 0.9850 | 0.9694 | 0.9858 |
| Information Fusion + Subpixel | **0.8533** | **0.8401** | **0.9852** | **0.9703** | **0.9888** |

**Table 2.** Ablation table for Chase dataset

| Method | F-1 | SE | SP | AC | AUC |
|---|---|---|---|---|---|
| CLAHE + Subpixel | 0.8576 | 0.8452 | 0.9878 | 0.9746 | 0.9912 |
| CLAHE + Ben Graham's + Subpixel | 0.8561 | 0.8370 | 0.9879 | 0.9740 | 0.9906 |
| Information Fusion + No-Subpixel | 0.8528 | 0.8308 | 0.9875 | 0.9736 | 0.9896 |
| Information Fusion + Subpixel | **0.8591** | **0.8472** | **0.9880** | **0.9747** | **0.9913** |

**Table 3.** Ablation table for Stare dataset

| Method | F-1 | SE | SP | AC | AUC |
|---|---|---|---|---|---|
| CLAHE + Subpixel | 0.8427 | 0.7842 | 0.9919 | 0.9721 | 0.9899 |
| CLAHE + Ben Graham's + Subpixel | 0.8682 | 0.8220 | 0.9925 | 0.9763 | 0.9945 |
| Information Fusion + No-Subpixel | 0.8677 | 0.8216 | 0.9924 | 0.9766 | 0.9941 |
| Information Fusion + Subpixel | **0.8686** | **0.8240** | **0.9926** | **0.9768** | **0.9945** |

## 4   Experimental Evaluation

This section provides the implementation details of the proposed method and a detailed analysis for a number of experiments performed for a robust evaluation of the method. The experiments are performed on a single 16GB NVIDIA Tesla P100 PCIe GPU. Instead of patch-based training we used full images resized to 512x512 resolution for both training and testing. A batch size of 5 is used, keeping in mind the hardware limitation. We used ADAM optimizer with an initial learning rate of 0.001. The learning rate is dynamically reduced by a factor 0.1 when the validation loss reaches a plateau. We used early stopping to stop the training when the validation loss remains stable for 10 consecutive epochs. Our model takes around an hour and a half on an average on the specified hardware to train on 1000 augmented full images of 512x512 resolution. Inference can be done within half a second on a 512x512 resolution image.

The results of the experiments and comparisons with other best approaches are provided in the following tables.

**Table 4.** Results

| Dataset | F-1 | SE | SP | AC | AUC |
|---------|-----|-----|-----|-----|-----|
| Drive | 0.8533 | 0.8401 | 0.9852 | 0.9703 | 0.9888 |
| Chase | 0.8591 | 0.8472 | 0.9880 | 0.9745 | 0.9913 |
| Stare | 0.8686 | 0.8240 | 0.9923 | 0.9762 | 0.9945 |

**Table 5.** Comparison on Drive dataset

| Method | Year | F-1 | SE | SP | AC | AUC |
|--------|------|-----|-----|-----|-----|-----|
| U-Net | 2018 | 0.8174 | 0.7822 | 0.9808 | 0.9555 | 0.9752 |
| Dense Block U-Net | 2018 | 0.8146 | 0.7928 | 0.9776 | 0.9541 | 0.9756 |
| DUNet | 2019 | 0.8237 | 0.7963 | 0.9800 | 0.9566 | 0.9802 |
| DE-UNet | 2019 | 0.8270 | 0.7940 | 0.9816 | 0.9567 | 0.9772 |
| Vessel-Net | 2019 | - | 0.8038 | 0.9802 | 0.9578 | 0.9821 |
| IterNet | 2020 | 0.8205 | 0.7735 | 0.9838 | 0.9573 | 0.9816 |
| Proposed Method | 2020 | **0.8533** | **0.8401** | **0.9852** | **0.9703** | **0.9888** |

**Table 6.** Comparison on Chase dataset

| Method | Year | F-1 | SE | SP | AC | AUC |
|--------|------|-----|-----|-----|-----|-----|
| U-Net | 2018 | 0.7993 | 0.7841 | 0.9823 | 0.9643 | 0.9812 |
| Dense Block U-Net | 2018 | 0.8006 | 0.8178 | 0.9775 | 0.9631 | 0.9826 |
| DUNet | 2019 | 0.7883 | 0.8155 | 0.9752 | 0.9610 | 0.9804 |
| DE-UNet | 2019 | 0.8037 | 0.8074 | 0.9821 | 0.9661 | 0.9812 |
| Vessel-Net | 2019 | - | 0.8132 | 0.9814 | 0.9661 | 0.9860 |
| IterNet | 2020 | 0.8073 | 0.7970 | 0.9823 | 0.9655 | 0.9851 |
| Proposed Method | 2020 | **0.8591** | **0.8472** | **0.9880** | **0.9745** | **0.9913** |

**Table 7.** Comparison on Stare dataset

| Method | Year | F-1 | SE | SP | AC | AUC |
|--------|------|-----|-----|-----|-----|-----|
| U-Net | 2018 | 0.7595 | 0.6681 | 0.9915 | 0.9639 | 0.9710 |
| Dense Block U-Net | 2018 | 0.7691 | 0.6807 | 0.9916 | 0.9651 | 0.9755 |
| DUNet | 2019 | 0.8143 | 0.7595 | 0.9878 | 0.9641 | 0.9832 |
| IterNet | 2020 | 0.8146 | 0.7715 | 0.9886 | 0.9701 | 0.9881 |
| Proposed Method | 2020 | **0.8686** | **0.8240** | **0.9923** | **0.9762** | **0.9945** |

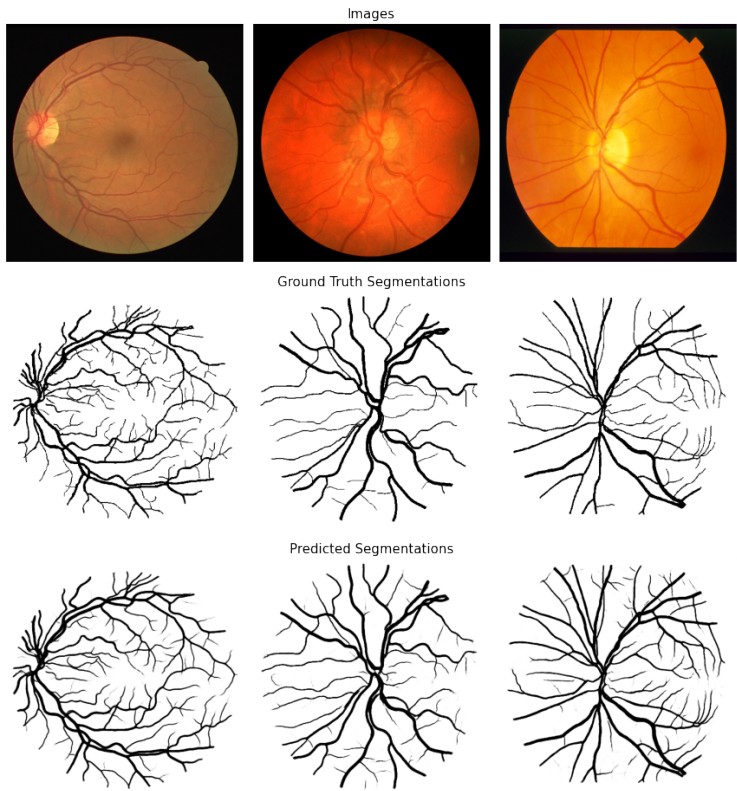

**Fig. 6.** From top to bottom: Original Images, Ground Truth Segmentations, Predicted Segmentations. From left to riht: Drive, Chase, Stare

## 5    Conclusion

In this experiment we showed that Subpixel Convolutions are very efficient in preserving information and are an effective way of changing image sizes during encoding and decoding of an image. Residual convolutional blocks can be adapted to use subpixel convolutions in place of max-pooling and thus a residual network can be improved to encode as much information as possible, both spatial and contextual with minimal loss of information during down-sampling. Similarly, subpixel based upsampling increases the spatial dimension of an image in a learnable way, preserving the information in an efficient manner. The proposed model SpruNet achieves AUC of 0.9888, 0.9913 and 0.9945 on the Drive, Chase and Stare datasets respectively. Thus beating the state-of-the-art models with a much simpler architecture with lesser parameter($\sim$20M) with a fast inference speed of 0.5 seconds on a 512x512 full image.

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
