# OpenReview forum: "A Subpixel Residual U-Net and Feature Fusion Preprocessing for Retinal Vessel Segmentation"
_thecvf.com/ECCV/2020/Workshop/BIC — BIC 2020 Oral_

### Official Review · AnonReviewer1 · 2020-07-30
**Easy to read paper with small but nice contributions, although comparison with other studies is limited.**

**Rating:** 6
**Confidence:** 4

**Review:**

### Summary
The authors propose an approach for the segmentation of retinal vessels. With their modifications, that they introduce in this paper (modified pre-processing, a combination of two losses, subpixel convolution modules for up-and downsampling in the U-Net), they achieve state-of-the art results (although interpretation of comparison with other studies is limited in my opinion). The new introduction of using subpixel convolutions also for down-sampling (it is usually used only for up-sampling) improves the method.


### Major strengths of the paper
- The paper is clear, results are presented nicely, and the paper is easy to understand.

- The authors try out different model parameters and present the results nicely in an ablation study, such that the reader can see the individual contributions.

- The results the authors achieve with their new method are better than other methods, but interpretation of results are limited (see next section). The authors use different modifications (such as pre-processing, loss combination, subpixel convolution) that improve the method, that are clearly presented such that it is easy to learn from this paper.

### Major weaknesses of the paper
- In this paper, the authors compare their results to results of other studies, but this comparison has several problems:
  1. Inference and Evaluation is done on a resized image (512x512) from sizes (584x565, 999x960, 700x605) for the three datasets. Evaluation on a downscaled ground-truth version makes it difficult to compare with other studies (this is not so relevant for the first or third dataset, but especially for the second dataset with a downscale factor of ~2). It is also unclear whether the downscaling of the ground-truth was carried out in a way to preserve thin vessels with only a 1-pixel diameter.
  2. In their results table, they provide results from other studies, but for two out of three datasets, there is no predefined train-test splitting (this is clearly not the author's fault but the benchmark is problematic). Instead, they use a random splitting meaning that their test-split likely differs and the results are only comparable in a very limited way. Only results on the Drive dataset are directly comparable.
  3. There is no validation set (but only training and test set), and it is unclear whether hyperparameter tuning was carried out on the training set or test set. They use their best performing model on the test set to compare with other models, which demonstrates some tuning on the test set. Overall, this seems to be a general problem of the community as the size of datasets that are available are limited and no clear & clean benchmark is set up.
  4. Given that results are good on all datasets and that all versions of their models achieve competitive results, it is still likely that results are indeed good, but caveats of the comparison should be addressed more clearly in the discussion part.

- Captions of the figures are too short (sometimes only a few words) which makes it difficult to understand the figures or to spot the relevant aspects of the figure.

### Detailed suggestions
- Figure 3 caption: Caption is too short, it would be nice to have a little walk-through in such a caption. Eg. guide the reader to the relevant things, mainly what are the contributions of the paper.

- Line 404: in the following tables → replace with the actual tables.

- Reference for DICE-loss is missing (in section 3.4).

### Language
"combined with Ben Graham’s [2] pre-procesing method of subtracting the local average colour"
Pre-procesing → pre-processing

Line 263 & 285: "In this the image" → In this image

Line 343: "vessel vs non-vesel pixels."
Non-vesel → non-vessel

Line 400: "around an hour and a half on an average"
Takes an hour and a half on average


**Reviews Visibility:**

I agree that my anonymized review is made publicly visible, if the submission is accepted.

---

### Official Review · AnonReviewer2 · 2020-07-31
**Marginal work leaving much to be desired**

**Rating:** 3
**Confidence:** 4

**Review:**

This paper focuses on a longstanding problem in medical image analysis: retinal vessel segmentation. The authors propose to apply a subpixel residual U-Net to segment the retinal vessels in preprocessed images fusing contrast enhancement and average color subtraction. Performance is evaluated on three public data sets.

In this reviewer's opinion, the work is out of scope, as the workshop is about biological image analysis rather than medical image analysis. But more importantly, the paper has too many shortcomings:

- Section 2 is a rather monotic enumeration of recent works in the field, one by one, without an overarching discussion of trends including solved aspects and remaining challenges. This lack of insights into the strengths and weaknesses of the discussed works makes it hard to appreciate the value of the proposed method or even why a new method is needed at all.

- Section 3.2 combines two well-known preprocessing methods (CLAHE and average color subtraction) and performs well-known data augmentation approaches.

- Sections 3.3-3.4 propose a deep neural network architecture that consists of well-known concepts (U-Net, ResNet, batch normalization, ReLU, subpixel convolution, BCE-Dice loss). It is not clear from the description what is really new, except for minor technical details, whose effects on the final results are not evaluated.

- Section 4 presents the training approach and the test results "as is", without any discussion of the experimental design or the findings, and the sensitivity of the latter to the system hyperparameters. Tables 1-3 show the results of an ablation experiment suggesting there is an advantage in using information fusion. Table 4 simply repeats the best results from Tables 1-3. A comparison with some of the methods listed in Section 2 is shown in Tables 5-7, suggesting the proposed method is superior, but no discussion is presented as to why the proposed method would be better. Nor is it clear that the differences in performance between any of the different methods is statistically significant.

- Section 5 states that the method outperforms "state-of-the-art models with a much simpler architecture with lesser parameter (~20M) with a fast inference speed of 0.5 seconds on a 512x512 full image". No numbers (parameters, speed) are given for the other models, so the claims are not substantiated.

All in all this paper is missing too much information to make it valuable to the community.

**Reviews Visibility:**

I agree that my anonymized review is made publicly visible, if the submission is accepted.

---

### Official Review · AnonReviewer3 · 2020-07-31
**Limited novelty, lack of discussion, but good results**

**Rating:** 6
**Confidence:** 3

**Review:**

This paper investigates a simple pre-processing method and a modified U-Net architecture for the task of retinal vessel segmentation. The pre-processing combines CLAHE and Ben Graham's algorithm. The modification to the U-Net consists in the use of subpixel convolutions in the down- and up-sampling paths. Results are presented on three public datasets and show consistently improved results over competing methods.

Quality and Clarity
-------------------------

The paper is generally well written and easy to follow. The related work section, however, would benefit greatly from a discussion of the differences to the proposed method.

Furthermore, the experimental evaluation lacks a detailed description of where scores of competing methods come from (did the authors run experiments themselves, or are they from other publications?) as well as a discussion of the results (only a table with all the numbers is given). I suggest the authors use the extra space they still have to expand this section.

Originality
-------------

The novelty of the proposed method is somewhat limited. None of the introduced methods is new (CLAHE and Ben Graham's pre-processing, subpixel convolutions). Nevertheless, the application of those methods to the task of retinal vessel segmentation is sensible and worth studying.

Significance
----------------

Results are presented on three datasets (Drive, Chase, and Stare) and compared against several competing methods. Due to non-standard splits of the Chase and Stare dataset (and no further information about whether the authors reimplemented competing methods), the results for those are hard to compare. On the Drive dataset, however, the method shows a consistent improvement in accuracy (under several metrics).

Pros
------
* largely well written
* convincing improvements on several datasets

Cons
-------
* unclear comparison to competing methods
* lack of discussion in related work and experimental results


**Reviews Visibility:**

I agree that my anonymized review is made publicly visible, if the submission is accepted.

---

### Decision · Program_Chairs · 2020-07-31

Accept (Oral)